# Influence of the Tikhonov Regularization Parameter on the Accuracy of the Inverse Problem in Electrocardiography

**DOI:** 10.3390/s23041841

**Published:** 2023-02-07

**Authors:** Tiantian Wang, Joël Karel, Pietro Bonizzi, Ralf L. M. Peeters

**Affiliations:** Department of Advanced Computing Sciences, Maastricht University, P.O. Box 616, 6200 MD Maastricht, The Netherlands

**Keywords:** electrocardiographic imaging, inverse problem, regularization parameter, zero-order Tikhonov regularization

## Abstract

The electrocardiogram (ECG) is the standard method in clinical practice to non-invasively analyze the electrical activity of the heart, from electrodes placed on the body’s surface. The ECG can provide a cardiologist with relevant information to assess the condition of the heart and the possible presence of cardiac pathology. Nonetheless, the global view of the heart’s electrical activity given by the ECG cannot provide fully detailed and localized information about abnormal electrical propagation patterns and corresponding substrates on the surface of the heart. Electrocardiographic imaging, also known as the inverse problem in electrocardiography, tries to overcome these limitations by non-invasively reconstructing the heart surface potentials, starting from the corresponding body surface potentials, and the geometry of the torso and the heart. This problem is ill-posed, and regularization techniques are needed to achieve a stable and accurate solution. The standard approach is to use zero-order Tikhonov regularization and the L-curve approach to choose the optimal value for the regularization parameter. However, different methods have been proposed for computing the optimal value of the regularization parameter. Moreover, regardless of the estimation method used, this may still lead to over-regularization or under-regularization. In order to gain a better understanding of the effects of the choice of regularization parameter value, in this study, we first focused on the regularization parameter itself, and investigated its influence on the accuracy of the reconstruction of heart surface potentials, by assessing the reconstruction accuracy with high-precision simultaneous heart and torso recordings from four dogs. For this, we analyzed a sufficiently large range of parameter values. Secondly, we evaluated the performance of five different methods for the estimation of the regularization parameter, also in view of the results of the first analysis. Thirdly, we investigated the effect of using a fixed value of the regularization parameter across all reconstructed beats. Accuracy was measured in terms of the quality of reconstruction of the heart surface potentials and estimation of the activation and recovery times, when compared with ground truth recordings from the experimental dog data. Results show that values of the regularization parameter in the range (0.01–0.03) provide the best accuracy, and that the three best-performing estimation methods (L-Curve, Zero-Crossing, and CRESO) give values in this range. Moreover, a fixed value of the regularization parameter could achieve very similar performance to the beat-specific parameter values calculated by the different estimation methods. These findings are relevant as they suggest that regularization parameter estimation methods may provide the accurate reconstruction of heart surface potentials only for specific ranges of regularization parameter values, and that using a fixed value of the regularization parameter may represent a valid alternative, especially when computational efficiency or consistency across time is required.

## 1. Introduction

In cardiology, the electrocardiogram (ECG) is a common and well-established method to measure the electrical activity of the heart, by placing electrodes on the body’s surface. However, even with extensive training, it may be difficult to use the ECG to directly determine the underlying mechanism of certain arrhythmia, or to localize the origin of an arrhythmic event [1]. Over the past few decades, electrocardiographic imaging (ECGI) has been developed as a more refined, non-invasive technology to enhance the conventional ECG, by providing more detailed numerical and visual information on the electrical functioning of the heart of a patient. ECGI aims at reconstructing the heart surface potentials, starting from the corresponding body surface potentials, and the geometry of the torso and the heart. Estimated heart surface potentials are subsequently used to visualize activation and recovery patterns on the epicardium, thereby facilitating the study of arrhythmogenic substrates and arrhythmia in patients [2]. At its core, ECGI involves solving an inverse problem, and the way in which this inverse problem is solved has an impact on the quality and the properties of the computed solution.

To formulate the ECGI inverse problem, we start from the *forward model* of electrocardiology, which assumes an instantaneous and linear relationship between the heart surface potentials $H_{t}$ and the corresponding body surface potentials $B_{t}$, at any given moment in time *t* [3]: (1)$$B_{t}=AH_{t}.$$

Here, $B_{t}$ is a vector of potentials at $N_{B}$ body surface electrodes and $H_{t}$ a vector of potentials at $N_{H}$ epicardial nodes (mesh points), both at time *t*. The $N_{B}\times N_{H}$ matrix *A* is a patient-specific transfer matrix that captures the geometry and conductivity relation between the heart and body surface (typically obtained from a CT or MRI scan) [4,5]. Commonly, both geometries are modeled to be static and *A* as time-invariant, and the torso volume conductor is assumed to be homogeneous. In this study, the matrix *A* is computed from the recorded patient’s geometry with software available from the SCIrun software repository [6], which uses an implementation of the Boundary Element Method (BEM) [7,8].

The *inverse problem* amounts to recovering the heart surface potentials $H_{t}$ from the recorded body surface potentials $B_{t}$ and an available estimate of the transfer matrix *A*. Due to attenuation and dispersion in the torso volume conductor, as well as the fact that the number of nodes $N_{H}$ used to create the heart mesh is generally larger than the number of body surface electrodes $N_{B}$, this inverse problem is ill-posed. [9,10] (An inverse problem is well-posed, in the sense of Hadamard, if and only if there exists a solution that is unique, and that depends continuously on the initial conditions [11]. It is called ill-posed otherwise.) A complicating and somewhat related issue is that the *solution algorithm* may also be ill-conditioned, making the computed solution for $H_{t}$ highly sensitive to small perturbations in the measurements $B_{t}$, to the extent of becoming inaccurate. Regularization techniques dealing with the ill-posedness and the solution’s sensitivity to noise and perturbations can be employed to stabilize (or regularize) the ECGI solution [10,11]. This involves the inclusion of a penalty term into the optimization procedure: (2)$${\hat{H}}_{t}=\arg\min_{H_{t}}\left(D\left(B_{t},AH_{t}\right)+\mu R\left(H_{t}\right)\right),$$
where *D* is a distance measure, *R* is the penalty operator, and μ≥0 is the regularization parameter. For *D*, usually, the squared distance induced by the $L_{2}$-norm is used, and we will do the same in this paper. For *R*, different choices have been employed in the literature, such as the squared $L_{2}$-norm (ridge regression or zero-order Tikhonov regularization) or the $L_{1}$-norm (lasso regression). Weighted norms are also an option. In first- or second-order Tikhonov regularization, the squared $L_{2}$-norm is applied to the first- or second-order spatial derivatives of $H_{t}$, instead of directly to $H_{t}$. Since differentiation is a linear operator, for discretized problems, this amounts to choosing a corresponding weighted norm. It is also possible to include a combination of several regularization terms, such as in elastic net regression, which convexly mixes the squared $L_{2}$-norm and the $L_{1}$-norm.

In this study, we will focus our attention on the influence of the regularization parameter on the accuracy of ECGI, when using zero-order Tikhonov regularization. Several methods have been proposed in the literature for the estimation of the optimal value of the regularization parameter μ. In [12], fourteen different regularization techniques were assessed and compared for the ECGI of patients with atrial fibrillation (AF) in a simulation study. Of the approaches not requiring invasive measurements, zero-order Tikhonov regularization performed similarly to more complex techniques. Five different methods for computing the regularization parameter μ were assessed in [13] in the context of solving the inverse problem through the Method of Fundamental Solutions, which were then compared with the results obtained via the BEM. It was found that the choice of the best regularization method and regularization parameter estimation method depend on the particular scenario being considered. In [14], the authors analyzed zero-order Tikhonov regularization and $L_{1}$-norm regularization, several different methods for computing the transfer matrix *A*, as well as five methods for choosing μ, all based on generalized singular value decomposition, both on simulated and experimental data. Similar to [13], they found that the choice of the best regularization parameter estimation method depends on the choice of the regularization method for solving the inverse problem and on whether the interest is in epicardial potential reconstruction or pacing site localization.

In addition to the different performance of the available methods, the ECGI reconstructions may also suffer from over-regularization [15] or under-regularization, regardless of the method used to compute an optimal value for the regularization parameter. In current practice, one often only experiences such effects while observing the reconstruction results, without a prior understanding of the potential effect of certain regularization parameter values on the solution of the inverse problem. At the same time, some regularization parameter estimation methods may display a lack of robustness, convergence, or efficacy in specific circumstances [16]. By turning the attention to the regularization parameter itself, it may be easier to observe the influence of different choices of parameter values on the reconstructed solutions, and consequently to assess the values provided by different regularization parameter estimation methods.

In this study, we take the viewpoint that in order to acquire more insight into the effects of the choice of regularization parameter μ, it makes sense to focus on the regularization parameter itself rather than to focus on comparing the performance of methods for choosing μ, as they all work well in some circumstances and less well in others. In line with these observations, the two objectives of this study were (A) to first investigate the influence of the regularization parameter on epicardial potential reconstruction accuracy (by focusing on zero-order Tikhonov regularization, and on a sufficiently large range of parameter values), and (B) to then analyze the performance of different regularization parameter estimation methods, in terms of the accuracy of the ECGI solution and in view of the results under (A). To achieve this, in this study, we examined the reconstruction accuracy with high-precision simultaneous heart (ground truth) and torso recordings from intact animals (instrumented anesthetized dogs) [17]. We analyzed three different ranges with varying order of magnitude for the regularization parameter, and five common methods of calculating an optimal value of the regularization parameter: (1) the L-Curve method [18,19], (2) the Generalized Cross-Validation (GCV) method [20], (3) the Composite Residual and Smoothing Operator (CRESO) method [21], (4) the Zero-Crossing (ZC) method [22], and (5) the U-Curve method [23]. Finally, we investigated the effect of using a fixed value of the regularization parameter on all cardiac beats. This is relevant, since a better understanding of the influence of a regularization parameter value on the accuracy of the reconstruction of epicardial potentials may help to assess which regularization parameter estimation methods can provide acceptable performance under which circumstances (or specific applications). Moreover, focusing on well-chosen fixed values of the regularization parameter may be desirable when computational efficiency or consistency across time (across multiple cardiac beats) is required.

The paper is structured as follows. In Section 2, we briefly introduce the inverse problem in electrocardiography, the selection of the parameter range to be investigated, and the five most common methods of estimating an optimal value of the regularization parameter. In Section 3, we describe the experimental data, and the experiments and analyses that we performed to achieve the objectives of this study. Results are presented in Section 4 and discussed in Section 5. Finally, we provide some conclusions in Section 6.

## 2. Materials and Methods

### 2.1. The Inverse Problem in Electrocardiography

Tikhonov regularization [24] is the most widely used regularization procedure in ECGI. It uses a sum of squares of errors to measure the quality of a solution, and trades it off against a squared $L_{2}$-norm penalty on the reconstructed epicardial potentials $H_{t}$ to promote the regularity of the solution: (3)$$\begin{array}{cc} {\hat{H}}_{t} & =\arg\min_{H_{t}}J\left(H_{t}\right)\\ J(H_{t})= & ∥AH_{t}-B_{t}{∥}_{2}^{2}+\lambda^{2}{∥\Gamma H_{t}∥}_{2}^{2}. \end{array}$$

For consistency with most of the ECGI literature, the regularization parameter μ is replaced by $\lambda^{2}$ (and with a slight abuse of terminology, λ is also referred to as a regularization parameter). The matrix Γ represents a regularization operator (unity, gradient, or Laplacian), and λ the regularization parameter. In this paper, zero-order Tikhonov regularization is used, also known as ridge regression, which means that Γ is chosen to be the identity matrix, giving preference to solutions with smaller (unweighted) norms [25].

The solution of (Equation 3) is obtained by setting the gradient of $J(H_{t})$ to zero [26], $\nabla J(H_{t})=0$, and solving the resulting square linear system of equations for $H_{t}$. Once λ is chosen, the solution of (Equation 3) is given by
(4)$${\hat{H}}_{t}={\left(A^{T}A+\lambda^{2}I\right)}^{-1}A^{T}B_{t}.$$

### 2.2. Choice of the Regularization Parameter λ

As mentioned, there exist different methods for estimating the regularization parameter value λ. Additionally, the reconstruction results can be sensitive to different values of λ. Using a large λ will lead to over-regularization, and overly smooth reconstruction results. A small λ, on the other hand, has little effect on the inverse problem, and the results may still be highly sensitive to noise. One can analyze the different approaches for computing the regularization parameter λ, and then dismiss those that consistently yield over-regularization or under-regularization. In this respect, it is important to perform a sensitivity analysis, in order to investigate how different values of λ can affect the inverse solution. In [17], the L-Curve method was employed on the same data as in this study in order to determine an optimal value for λ. Since λ is computed per time instant during one heart beat, but a certain degree of coherence between the reconstructions within a single beat is generally desirable (e.g., to compute activation or repolarization times), the median λ generated by the L-Curve method among all time instants per single beat was used in [17]. The values encountered for λ were around 0.01 for all heart beats. Following up on this result, we decided to focus our sensitivity analysis of λ on the following three consecutive intervals: 0.001–0.009 (low-regularization; stepsize: 0.001), 0.01–0.09 (medium-regularization; stepsize: 0.01), 0.1–1.0 (high-regularization; stepsize: 0.1).

Additionally, we analyzed five different methods for estimating the regularization parameter λ at a given time instant, namely (1) the L-Curve method [18,19], (2) the Generalized Cross-Validation (GCV) method [20], (3) the Composite Residual and Smoothing Operator (CRESO) method [21], (4) the Zero-Crossing (ZC) method [22], and (5) the U-Curve method [23].

#### 2.2.1. L-Curve Method

The L-Curve method [18,19] is the most popular method for estimating the regularization parameter λ in ill-posed problems [26]. It involves a log-log plot of the regularized solution norm versus the corresponding residual norm for all valid regularization parameters: (5)$$L\left(\lambda \right)=\{(∥A{\hat{H}}_{t,\lambda }-B_{t}{∥}_{2},∥{\hat{H}}_{t,\lambda }{∥}_{2}),\lambda >0\}$$

This plot is expected to show an L-shape (referred to as the L-Curve). The point of maximum curvature of the L-Curve (known as the “L-corner”) represents a trade-off between the solution norm and residual norm and is selected to be an optimal value for λ [27]. Figure 1a shows an example of an L-Curve plot, with the L-corner of the plot indicated, and the corresponding value λ=0.035874.

#### 2.2.2. Generalized Cross-Validation Method

The Generalized Cross-Validation (GCV) method [20] is a popular method for estimating the regularization parameter in problems with discrete data and stochastic noise [28]. The regularization parameter λ is estimated by minimizing the weighted prediction errors G(λ) when using all “leave-one-out” (ordinary cross-validation) regularized solutions [29,30]: (6)$$G\left(\lambda \right)=\frac{∥\left(I-A\left(\lambda \right)\right)B_{t}{∥}_{2}^{2}}{{\left[\frac{1}{N_{B}}\mathrm{Tr}\left(I-A\left(\lambda \right)\right)\right]}^{2}}$$
with $A\left(\lambda \right)=A{\left(A^{T}A+\lambda^{2}I\right)}^{-1}A^{T}$. According to Equation (Equation 4), Equation (Equation 6) can also be written as
(7)$$G\left(\lambda \right)=\frac{∥A{\hat{H}}_{t,\lambda }-B_{t}{∥}_{2}^{2}}{{\left[\frac{1}{N_{B}}\mathrm{Tr}\left(I-A\left(\lambda \right)\right)\right]}^{2}}$$

With singular value decomposition applied to matrix *A*, $A=U\Sigma V^{T}$, the trace term is estimated by
(8)$$\mathrm{Tr}\left(I-A\left(\lambda \right)\right)=N_{B}-r+\sum_{i=1}^{r}\frac{\lambda^{2}}{\sigma_{i}^{2}+\lambda^{2}}.$$

Here, *r* is the rank of transfer matrix *A*, and $\sigma_{i}$ is the *i*-th singular value of *A* arising from diagonal matrix Σ with $\sigma_{i}$ arranged in descending order. Thus, G(λ) can be expressed as
(9)$$G\left(\lambda \right)=\frac{\sum_{i=1}^{r}\frac{\lambda^{4}{\left(u_{i}^{T}B_{t}\right)}^{2}}{{\left(\sigma_{i}^{2}+\lambda^{2}\right)}^{2}}+\sum_{i=r+1}^{N_{B}}{\left(u_{i}^{T}B_{t}\right)}^{2}}{{\left[\frac{1}{N_{B}}\left(N_{B}-r+\sum_{i=1}^{r}\frac{\lambda^{2}}{\sigma_{i}^{2}+\lambda^{2}}\right)\right]}^{2}},$$
where $u_{i}$ is the *i*-th column of orthogonal matrix *U*. Figure 1b shows an example of a G(λ) curve, also indicating the estimated value $\lambda =2.5119\times 10^{-10}$ corresponding to the minimum of G(λ).

#### 2.2.3. Composite Residual and Smoothing Operator Method

In the Composite Residual and Smoothing Operator (CRESO) method [21], λ>0 is chosen to be the value corresponding to the first local maximum of C(λ), defined as the difference between the derivative of the solution squared norm and the derivative of the residual squared norm C(λ) [26]: (10)$$C\left(\lambda \right)=\frac{d}{d\lambda^{2}}\left(\lambda^{2}∥{\hat{H}}_{t,\lambda }{∥}_{2}^{2}-{∥A{\hat{H}}_{t,\lambda }-B_{t}∥}_{2}^{2}\right).$$

However, to boost performance, we deviated from the standard in other references and decided to adopt the last local maximum in this paper, since our experiments showed that it achieves considerably better reconstruction results than the first local maximum in our application. Figure 1c shows an example of a C(λ) curve, indicating the estimated value λ=0.019801 corresponding to the last local maximum of C(λ).

#### 2.2.4. Zero-Crossing Method

The Zero-Crossing (ZC) method [22] chooses λ to be the smallest value such that the residual squared norm and the contribution of the penalty term to the criterion are both equal: (11)$$B\left(\lambda \right):=\lambda^{2}∥{\hat{H}}_{t,\lambda }{∥}_{2}^{2}-{∥A{\hat{H}}_{t,\lambda }-B_{t}∥}_{2}^{2}=0.$$

Figure 1d shows an example of a B(λ) curve, together with the estimated value λ=0.067403 corresponding to the first point where B(λ)=0.

#### 2.2.5. U-Curve Method

The U-Curve [23] is defined as the plot of the sum of the reciprocals of the regularized solution squared norm and of the residual squared norm [23]: (12)$$U\left(\lambda \right)=\frac{1}{∥A{\hat{H}}_{t,\lambda }-B_{t}{∥}_{2}^{2}}+\frac{1}{∥{\hat{H}}_{t,\lambda }{∥}_{2}^{2}}$$

The U-Curve is expected to have a U-shape, hence its name. The optimal λ is located where U(λ) achieves its minimum value, while the sides of the curve correspond to regularization parameters for which either the solution norm or the residual norm dominates. Figure 1e shows an example of a U(λ) curve in which λ=0.37471 corresponds to the minimum of U(λ). A log-log plot is commonly used to display the U-shape clearly, as in Figure 1e.

## 3. Analyses

### 3.1. Data

In this study, we used the same data set as in [17]. These experimental data were obtained from four healthy anesthetized dogs. Unipolar potential recordings were obtained through epicardial electrodes (with sampling frequency 1000 Hz). Two silicone bands (99 electrodes) were implanted around the basal and mid-basal ventricular epicardium, and additional electrodes were placed at the LV apical epicardium, the LV endocardium, the RV apical endocardium, and the right atrial endocardium (103 electrodes in total) after thoracotomy. Simultaneous body surface potential recordings were obtained from body surface electrodes attached to the chest (total number varying from 184 to 216, depending on torso size; with sampling frequency 2048 Hz) [17]. Using a CT scan, the torso–heart geometry was digitized, including the body surface electrodes, the implanted electrodes, and the ventricular epicardial surface (on average 1693 mesh nodes, mean node-to-node distance 4 mm). A total of 92 different beats were recorded from the four dogs, including 6 sinus beats and 86 paced beats (only one beat included for each pacing site). On average, 60 epicardial electrodes provided high-quality potential recordings [17]. After reconstruction of the electrograms for all 92 beats, this resulted in 5488 pairs of recorded and reconstructed electrograms for single beats. The (local) activation and recovery times were computed as in [17].

### 3.2. Predefined Interval

For the five previously discussed methods, before determining the optimal regularization parameter value λ, a range in which λ varies should be selected first. This choice of interval for λ variation has an impact, since these methods for computing the optimal regularization parameter are entirely data-driven and may not return a value for λ if the interval is too narrow, or an undesirably small λ if the interval is too wide. In our experiments, the predefined interval is set, by means of the singular values of the transfer matrix *A*, to be $\left[\sigma_{\mathrm{round}(r/2)},\sigma_{1}\right]$. For the four dogs, the corresponding intervals are $\left[5.1\times 10^{-7},0.2470\right]$, $\left[6.1\times 10^{-5},0.3457\right]$, $\left[3.6\times 10^{-5},0.3429\right]$, and $\left[1.2\times 10^{-6},0.4383\right]$, which appear to work suitably well. The interval for the GCV method is chosen as $\left[1\times 10^{-30},\sigma_{1}\right]$. For the U-Curve method, the wider interval $\left[\sigma_{\mathrm{round}(r/2)},4.44\right]$ is used.

### 3.3. Statistical Analysis

For each epicardial electrode and each cardiac beat, the accuracy of the ECGI reconstruction was assessed by comparing the recorded and the reconstructed potentials at the corresponding (closest) virtual epicardial node. Performance was measured in terms of the relative root-mean-squared error (RE) and the Pearson correlation coefficient (CC). Both recorded and reconstructed epicardial potentials were normalized before computing the RE, by means of Min–Max normalization, and the electrograms were aligned to maximize their cross-correlation. The RE and the CC were also computed for activation times and recovery times, estimated through the recorded and reconstructed potentials.

### 3.4. Analyses

#### 3.4.1. Influence of the Regularization Parameter on the ECGI Inverse Solution

The influence of the regularization parameter λ on the ECGI inverse solution was analyzed by measuring the CC and the RE for epicardial potentials, activation times, and recovery times, for all values of λ in the ranges defined in Section 2.2—namely, 0.001–0.009 (low-regularization; step: 0.001), 0.01–0.09 (medium-regularization; step: 0.01), 0.1–1.0 (high-regularization; step: 0.1). For each beat, different time instants can yield different λ values. Therefore, for the reconstruction of a beat, we used the median λ from all time instants of that beat (as in [17]).

#### 3.4.2. Performance of Different Regularization Parameter Estimation Methods

As already mentioned in Section 1 and Section 2.2, we analyzed the performance of five different regularization parameter estimation methods, in terms of the accuracy of the ECGI solution, and in view of the results of the previous analysis.

#### 3.4.3. Effect of Using a Fixed Value of λ for All Beats

To understand the effect of using a fixed value of λ on all beats, we used a fixed λ for all beats in a dog, selected from the optimal λ-range obtained from the first analysis.

## 4. Results

### 4.1. Influence of the Regularization Parameter on the ECGI Inverse Solution

Figure 2 shows the boxplots of the CC and the RE between the recorded and the reconstructed electrograms (two top panels), the measured and reconstructed activation times (two middle panels), and the measured and the reconstructed recovery times (two bottom panels), for all values of λ in the ranges 0.001–0.009, 0.01–0.09, and 0.1–1.0. In general, it can be noticed that the mid-regularization range provides larger CC and smaller RE values for reconstructed potentials and activation/recovery times. The best results are achieved for λ∈ [0.02, 0.1], [0.01, 0.03], [0.02, 0.03], respectively (with λ= 0.02 included in all three intervals).

### 4.2. Performance of Different Regularization Parameter Estimation Methods

Figure 3 shows the λ values provided by all five methods analyzed in this study (L-Curve, GCV, CRESO, ZC, and U-Curve), computed on the 92 beats available from the four dogs. Figure 3a shows the scatter plots of λ values for all beats and each dog, and Figure 3b–f show the corresponding distributions for the different methods. L-Curve, CRESO, and ZC show comparable distributions, with median values around 0.01, which lies in the medium-regularization range. The median for GCV was almost 0, with only two beats yielding a λ greater than 0.01. The median for the U-Curve was 0.3, thus showing a preference for a higher regularization range.

Figure 4 shows the CC and RE for epicardial electrograms (top), activation times (middle), and recovery times (bottom) for all five methods analyzed in this study.

In general, GCV gave the smallest CCs and largest REs for reconstructed electrograms and activation/recovery times, followed by the U-Curve. L-Curve, CRESO, and ZC showed similar performance, and they performed better than the other two methods in terms of reconstructed potentials and activation/recovery times.

### 4.3. Effect of Using a Fixed Value of λ for All Beats

The results of Section 4.2 suggest that L-Curve, CRESO, and ZC can achieve better results than GCV and U-Curve in terms of reconstructed electrograms and activation/recovery times. The median values of λ for these three methods are around 0.01. This is consistent with the results presented in Section 4.1, which showed that a value of λ=0.01 can give high performance in terms of reconstructed electrograms and activation/recovery times, compared to other λ values analyzed in this study. Consequently, we analyzed the effect of using a fixed value of λ=0.01 for all 92 beats. Figure 5a–c show the distributions of CC and RE for electrograms, activation times, and recovery times for L-Curve, CRESO, and ZC in comparison to the distributions of CC and RE when a fixed value of λ=0.01 is used for all 92 beats. It can be noticed that using a fixed value of λ=0.01 for all beats provides comparable results to the performance of L-Curve, CRESO, and ZC, despite the fact that these three methods compute an optimal value of λ for each beat. Figure 5d depicts the relationship between electrograms, activation times, and recovery times, estimated with either L-Curve, CRESO, or ZC, vs. a fixed value of λ=0.01. It can be noticed that the points are generally well concentrated around the identity line, thus confirming that the results of L-Curve, CRESO, and ZC are very similar to the results obtained with a fixed λ=0.01.

Figure 6 shows the activation time maps of a beat for which L-Curve, CRESO, and ZC achieved the highest CC, together with the corresponding maps when a fixed value of λ equal either to 0.009, 0.01, 0.02, or 0.03 is used, for comparison. It can be noticed that both λ=0.01 and 0.02 achieve the highest CC among the four fixed λ values. Results also show that the CC achieved with λ=0.01 is comparable to those achieved with L-Curve, CRESO, and ZC, and the activation maps are very similar upon visual inspection. Moreover, all three methods achieve the highest CC for λ values very close to 0.01, thus supporting the evidence of this being a suitable value to be chosen for the zero-order Tikhonov regularization of the inverse problem in electrocardiography.

## 5. Discussion

In this study, we analyzed the influence of the regularization parameter on the accuracy of the inverse problem in electrocardiography when using zero-order Tikhonov regularization. Importantly, we took the viewpoint that in order to acquire more insight into the effects of the choice of regularization parameter λ, it makes sense to focus on the regularization parameter itself, rather than focusing on comparing the performance of methods for choosing λ. For this purpose, we analyzed a sufficiently large range of parameter values, including low-regularization (0.001–0.009), medium-regularization (0.01–0.09), and high-regularization (0.1–1). We analyzed the performance in terms of the reconstruction of epicardial electrograms and estimation of activation and recovery times, by comparing them with the ground truth available from the epicardial electrodes. Results showed that medium regularization provides the best performance in general, with the λ range (0.01–0.03) showing high performance for electrograms, activation times, and recovery times (and with λ=0.02 achieving the best performance). This is also in line with the current results that we obtained on patients, for which optimal λ values around 0.01 were also observed in human data (unpublished results).

We then analyzed the performance of five different regularization parameter estimation methods (L-Curve, Generalized Cross-Validation (GCV), Composite Residual and Smoothing Operator (CRESO), Zero-Crossing (ZC), and U-Curve), in terms of the accuracy of the ECGI solution. All methods estimate a λ value for each beat. L-Curve, CRESO, and ZC showed the best performance in terms of the reconstruction of epicardial electrograms and estimation of activation and recovery times. Additionally, all three methods estimated λ values in the medium-regularization range (shown to be the best-performing range in our first analysis), with a median λ value of around 0.01. U-Curve showed lower performance and estimates of λ in the high-regularization range (median value of around 0.3). GCV showed the lowest performance, with estimates of λ values very close to zero. We assume that this may be due to the fact that GCV is designed for large-scale problems and performs well when the number of mesh points approaches infinity. Therefore, for small or medium-scale problems, it can provide very small values of λ, which renders the solution too under-regularized (as shown in our results, and also in [14]). Similar results to ours were reported in [31], in which L-Curve, ZC, and CRESO also showed similar performance. Contrary to our results, in [26,32], it was found that the U-Curve performed better than the L-Curve, which gave over-regularized solutions. This may be due to the fact that the Method of Fundamental Solutions, instead of the Boundary Element Method, was used in their study to solve the forward problem.

Finally, we analyzed the effect of using a fixed value of λ on all beats, and compared the performance with that of L-Curve, ZC, and CRESO. For this, we chose a value of λ=0.01, this being the median value estimated by the three methods above, and also a value within the optimal range identified by our first analysis (0.01–0.03). In general, our results showed that for the data set used in this study, using a fixed λ of 0.01 could provide comparable performance to L-Curve, ZC, and CRESO, despite this being applied to different dogs, with different geometries (number of mesh points 3276, 1321, 1064, and 1114, respectively, for the four dogs). This finding shows that using a fixed value of λ=0.01 when solving the inverse problem in electrocardiography with zero-order Tikhonov regularization may be sufficient to provide acceptable accuracy of the solution, in terms of reconstructed epicardial electrograms and estimated activation and recovery times. Additionally, this finding suggests that the ECGI solution can show a certain degree of stability across beats. This likely depends on specific conditions, and we may wonder whether well-chosen values of λ can be selected for specific situations. This could be an advantage in problems where coherence across time is desirable. When applying the CRESO method, we used the last local maximum rather than the first. The reason for this choice is that we noticed that for most time instants of each beat, only two local maxima were generally present in the C(λ) curve, with the first local maximum giving a λ value around $10^{-4}$ for almost all time instants, thus rendering the solution under-regularized. However, using the second local maximum gave a median λ around 0.01 for all 92 beats. This is consistent with the results reported in [31], where the second local maximum of the CRESO method was also used, and failure in using the first local maximum was attributed to geometry noise in the epicardial geometry.

With the increasing importance in healthcare of cost-effective, continuous, easy to deploy, and automated data analysis aimed at detecting a disease at an early stage and at improving patient stratification, we believe that the results of this study represent a relevant contribution to future remote monitoring ECGI solutions.

In this study, we only included data from animals, to have ground truth available about epicardial potentials and be able to assess the accuracy of reconstruction. As the data came from four different dogs with different geometries, we consider our findings to be sufficiently robust, and we believe that they can provide an indication of what we could expect in humans as well. As in [17], we only analyzed sinus rhythms and paced beats, to obtain a clear understanding of the influence of the regularization parameter on ECGI accuracy. We did not investigate any arrhythmia in this study, which is a much more complex phenomenon and difficult to reconstruct, thus potentially making it more challenging to achieve the objectives of this study. To what extent the findings of this study extend to cardiac arrhythmia is an aspect that needs to be investigated in the future.

## 6. Conclusions

In this study, we investigated the influence of the regularization parameter on the accuracy of the reconstruction of epicardial potentials, by assessing the reconstruction accuracy with high-precision simultaneous heart and torso recordings from intact animals. Results showed that values of the regularization parameter in the range (0.01–0.03) provide the best accuracy, and the three best-performing estimation methods (L-Curve, Zero-Crossing, and CRESO) provided values in this range. Moreover, when using a fixed value of the regularization parameter, we could achieve comparable performance to the three best methods. This suggests that using a well-chosen fixed value of the regularization parameter with zero-order Tikhonov regularization may be sufficient to provide an acceptable solution to the inverse problem in electrocardiography. These findings are relevant as they provide insights into how to assess results from regularization parameter estimation methods in the context of the inverse problem in electrocardiography, and may help to select a suitable regularization parameter estimation method for a specific application. Moreover, focusing on well-chosen fixed values of the regularization parameter may be an advantage when computational efficiency or consistency across time (across cardiac beats) is desirable.

## Figures and Tables

**Figure 1 sensors-23-01841-f001:**
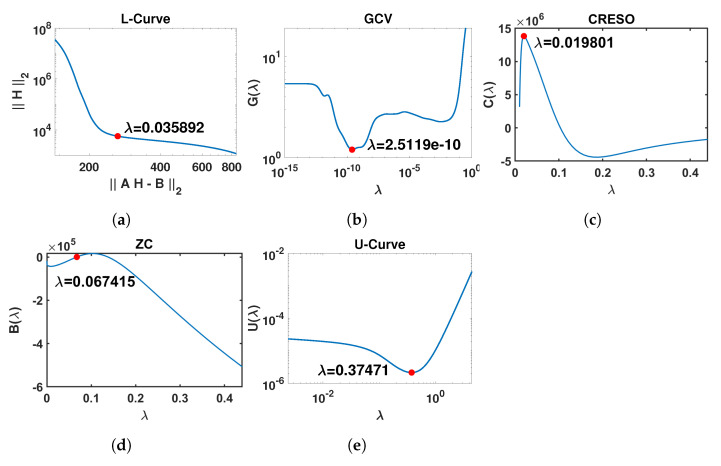
Graphical illustration of the outputs of the five methods included in this study for the estimation of the regularization parameter λ. The red points represent the optimal value of λ for each method. In order to present the different curves clearly, only part of the x-axis is shown. (**a**) L-Curve: log-log plot of the regularized solution norm ${∥H∥}_{2}$ versus the corresponding residual norm ${∥AH-B∥}_{2}$ across the full range for $\lambda \in [10^{-6},0.44]$. (**b**) GCV: G(λ) curve for $\lambda \in [10^{-15},0.44]$. (**c**) CRESO: C(λ) curve for $\lambda \in [10^{-2},0.44]$. (**d**) ZC: B(λ) curve for $\lambda \in [10^{-6},0.44]$. (**e**) U-Curve: log-log plot of the U(λ) curve for $\lambda \in [10^{-3},4.44]$.

**Figure 2 sensors-23-01841-f002:**
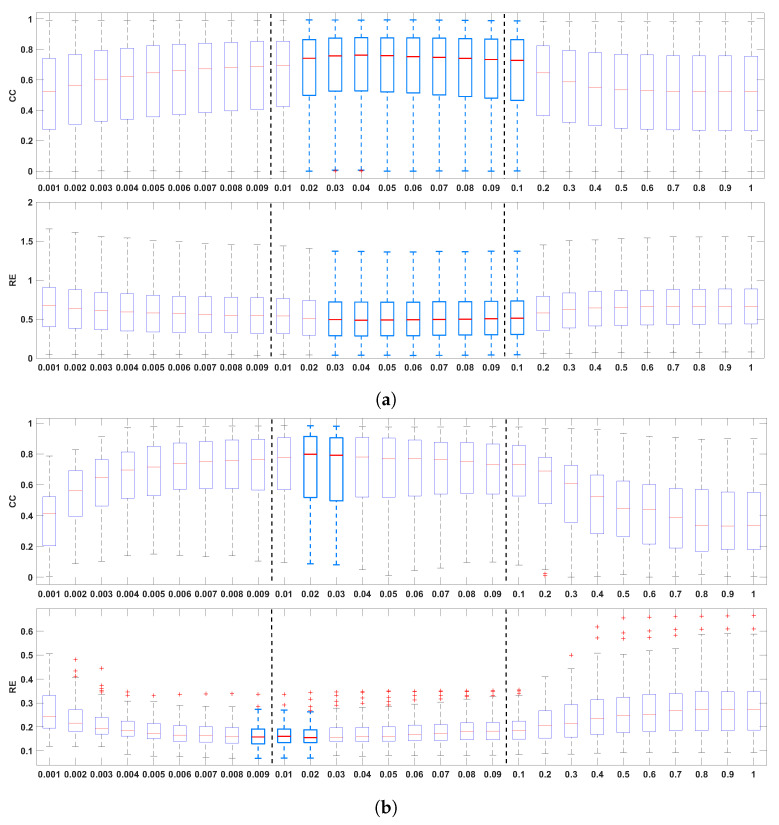
Boxplots summarizing the influence of the regularization parameter on reconstructed electrograms ((**a**), two top panels), activation times ((**b**), two middle panels), and recovery times ((**c**) two bottom panels). The best results are indicated with light blue boxplots. Whiskers range from the end of the interquartile range to the furthest observation within the whisker length that is not considered an outlier.

**Figure 3 sensors-23-01841-f003:**
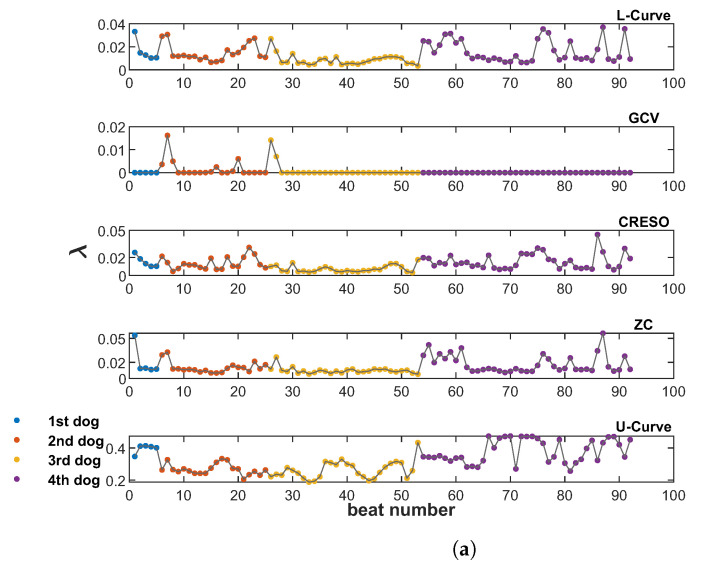
Scatter plots (**a**) and violin plots (**b**–**f**) of λ values estimated by all five methods analyzed in this study, and for all 92 beats from the four dogs. (**a**): Scatter plot of λ values from the five methods included in this study, for all 92 beats from the four dogs (from top to bottom: L-Curve, GCV, CRESO, ZC, U-Curve. (**b**): Violin plot of λ for L-Curve. (**c**): plot of λ for GCV. (**d**): plot of λ for CRESO. (**e**): plot of λ for ZC. (**f**): Violin plot of λ for U-Curve.

**Figure 4 sensors-23-01841-f004:**
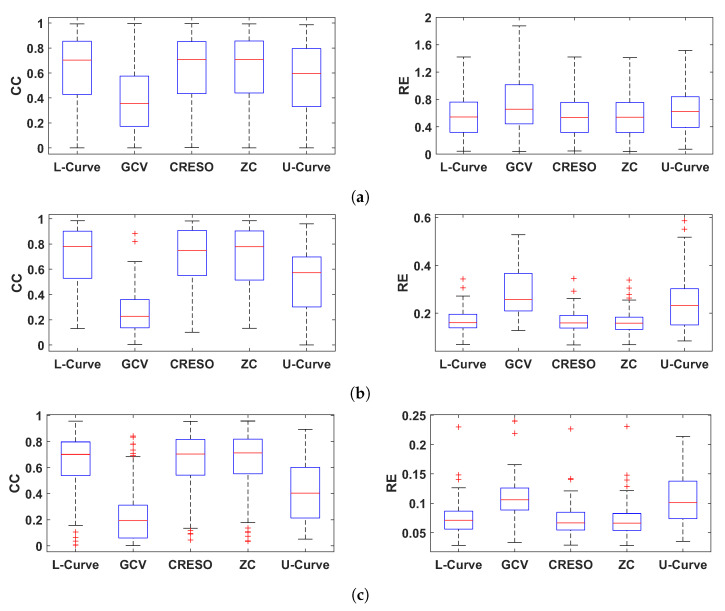
Boxplots summarizing the performance of the five methods for estimating the regularization parameter analyzed in this study. Performance is shown in terms of CC and RE for reconstructed electrograms (**a**), activation times (**b**), and recovery times (**c**).

**Figure 5 sensors-23-01841-f005:**
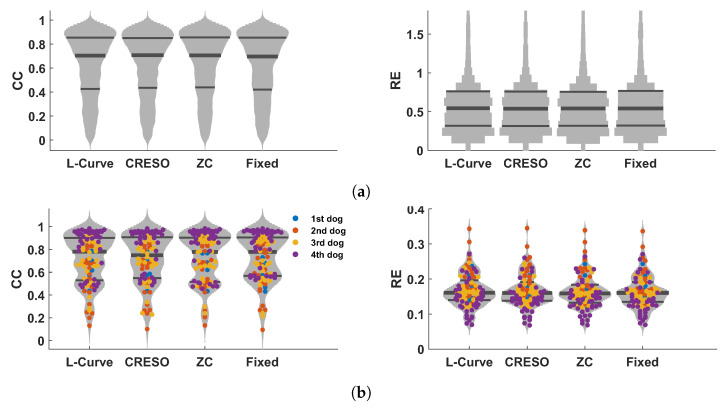
Results summarizing the performance comparison between L-Curve, CRESO, and ZC vs. when using a fixed value of λ=0.01 for all 92 beats used in this study. (**a**): Violin plots of CC (left) and RE (right) between measured and reconstructed electrograms for L-Curve, CRESO, ZC, and when using a fixed value of λ=0.01 for all 92 beats. Only the distributions are provided in this plot, because of the large amount of time instants considered for the electrograms (5488). (**b**): Violin plots of CC (left) and RE (right) between measured and reconstructed activation times for L-Curve, CRESO, ZC, and when using a fixed value of λ=0.01 for all 92 beats. (**c**): Violin plots of CC (left) and RE (right) between measured and reconstructed recovery times for L-Curve, CRESO, ZC, and when using a fixed value of λ=0.01 for all 92 beats. (**d**): Upper-left: relation between activation times (AT) estimated with either L-Curve, CRESO, or ZC vs. when using a fixed value of λ=0.01. Upper-right: similar, but for recovery times (RT). Lower-left: similar, but for electrograms.

**Figure 6 sensors-23-01841-f006:**
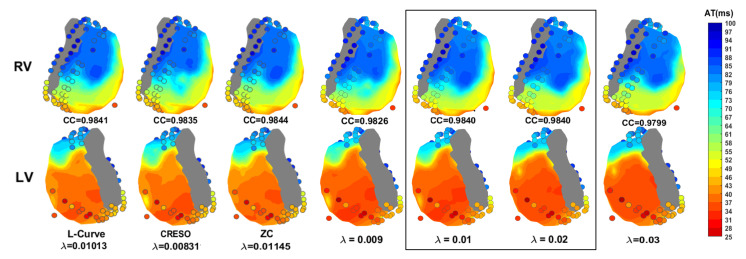
Activation time maps of a beat from the fourth dog for which L-Curve, CRESO, and ZC achieve the highest CC, together with the corresponding maps when a fixed value of λ equal either to 0.009, 0.01, 0.02, or 0.03 is used. RV: right ventricle. LV: left ventricle.

## Data Availability

The data set was provided by Matthijs Cluitmans. Part of the data is available on https://www.ecg-imaging.org/edgar-database.

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
