# Peer review of "Influence of the Tikhonov Regularization Parameter on the Accuracy of the Inverse Problem in Electrocardiography"

_sensors, 2023, doi:10.3390/s23041841_

Round 1

Reviewer 1 Report

This investigates the influence of the regularization parameter on the accuracy of the reconstruction of heart surface potentials, by assessing reconstruction accuracy with high-precision simultaneous heart and torso recordings from four dogs. This research has certain significance, but at present there are the following shortcomings.

1.     Several grammar errors/typo/spell errors throughout, need a critical language check. Most importantly, irregular usage of uppercase letters in the mid of the sentences

2.     The font of the symbols in the formula needs to be carefully checked and kept consistent.

3.       What is the significance of this work? What are the potential applications?

Reviewer 2 Report

1. Paper focus on an interesting topic of study.

2. Organization of the paper is good and accepted.

3. Theoretical background, basics, equations are well explained.

4. Comparative results presented validate the proposed framework.

5. The novelty of the paper is sound which is highlighted with good number of references. However, I would further elaborate on typical applications and improve the motivation in the abstract and introduction. Currently, the paper is "dry" and lacks a wider context.

6. References are up to date and properly cited as well.
